# Parkinson’s Disease in Louisiana, 1999–2012: Based on Hospital Primary Discharge Diagnoses, Incidence, and Risk in Relation to Local Agricultural Crops, Pesticides, and Aquifer Recharge

**DOI:** 10.3390/ijerph17051584

**Published:** 2020-02-29

**Authors:** Martin E. Hugh-Jones, R. Hampton Peele, Vincent L. Wilson

**Affiliations:** 1Department of Environmental Sciences, College of the Coast & Environment, Louisiana State University, Baton Rouge, LA 70803, USA; monster77@aol.com; 2Cartographic Section, Louisiana Geological Survey, Louisiana State University, Baton Rouge, LA 70803, USA; hampton@lsu.edu

**Keywords:** agricultural pesticides, 2,4-D, chlorpyrifos, paraquat, aquifer recharge, woodland, pastures, agricultural ecology

## Abstract

The two major causes of Parkinson’s disease (PD) are genetic susceptibility and exposure to agricultural pesticides. Access to 23,224 individuals’ hospital primary discharge diagnoses of PD allowed the mapping of cases against known crop distributions and pesticides. Our main objective was to map PD risks (cases per 10,000 people) against crops and their pesticides. The ZIP Code address locations, and the 2000 and 2010 census data, were used to map the risks of PD throughout Louisiana and in relation to United States Department of Agriculture (USDA)-recorded crops. The introduction of glyphosate-resistant crops appears to have initiated the PD disappearance from northeastern parishes on the west bank of the Mississippi river. Rice and sugar cane are seemingly unassociated with PD, as is the Mississippi itself, except for Jefferson and St. Charles Parishes, which are essentially urban or industrial. The present major PD-affected areas are associated with commercial forests, woodlands, and pastures, and thus with certain arbor-pastoral pesticides, 2,4-D, chlorpyrifos, and paraquat. Human populations at maximum risk are those living in areas of moderate and high aquifer-recharge potential. The levels of estimated pesticide exposure in these recharge areas we were able to access were of variable use, but significant amounts of 2,4-D were being used.

## 1. Introduction

Parkinson’s disease (PD) is a common neurodegenerative disease, second worldwide only to Alzheimer’s disease. Some one million Americans live with PD, which is more than the combined total of people diagnosed with multiple sclerosis, muscular dystrophy, and amyotrophic lateral sclerosis. Some 60,000 new cases are diagnosed each year [1]. PD is the consequence of dopaminergic neuron degeneration in the substantia nigra associated with the presence in the neurons of large ɑ-synuclein fibril eosinophilic inclusions, known as Lewy bodies. Clinical PD is characterized by chronic progressive tremor, bradykinesia, rigidity, and postural instability, but this becomes apparent only after a loss of 70%–80% of dopaminergic neurons [2]. This can result in an asymptomatic latent period of some 6 years [3]; some authors have suggested this pre-symptomatic phase may start early in life, and developmental insults (e.g., early pesticide exposure [4] and intrauterine infections [5]) could contribute to PD later in life.

Damage to the neurons may begin up to two decades before clinical symptoms begin. This early damage may be expressed in a partial or total loss of the ability to smell (hyposmia, anosmia); GI abnormalities, such as constipation, which can precede motor defects by many years; and sleepwalking and acting out of dreams, which may be unusually vivid (Rapid Eye Movement sleep behavior disorder or RBD) [6,7,8,9]. The latter serve as early markers for PD. Heiko Braak, a clinical neuroanatomist at Goethe University in Frankfurt, Germany, has proposed that PD proceeds in six cumulative stages, from non-motor to motor symptoms that correlate with the distribution of Lewy bodies in the nervous system [10]. Stage 1 involves lesions in the medulla oblongata, which would explain the olfactory symptoms. Stage 2 involves medulla oblongata and pontine tegmentum with sleep disorders. Stage 3 involves the pathology of Stage 2 plus midbrain lesions, including the substantia nigra and initial motor symptoms. Stage 4 involves the pathology of Stage 3 plus lesions in the basal prosencephalon and mesocortex, with increased motor symptoms. Stages 5 and 6 affect the neocortex with emotional and cognitive impairment and eventually dementia. Degeneration of the nigrostriatal dopaminergic neurons is a hallmark of PD, with clinical symptoms manifesting when some 50%–60% of these neurons are lost [11]. In summary, while late stage PD may be readily diagnosed, the earlier the stage, the more uncertain the recognition, much less the diagnosis. Unfortunately, with modern image technology, PD has tended to be over-diagnosed. A 1998 study of pre-routine brain scans found a clinical diagnostic specificity of 92%, a sensitivity of 80%, and a positive predictive value of only 46% [12].

### 1.1. Epidemiology

Some 85%–90% of PD cases are sporadic, with only 10%–15% of patients reporting a family history of the disease. In relation to the latter, early onset cases have a higher concordance rate in monozygotic twins than dizygotic twins. In general, exposure to specific environmental agents seems to be an important component of PD pathogenesis.

Normally, there are few cases before the age of 40, and the incidence increases rapidly after age 60. The incidence is commonly some 1.5 to 2.0 times higher in men than women. Women tend to develop the disease some 2 years later than men and have greater motor skills than PD-affected men. Part of this protection may be due to higher estrogen levels; it is relevant that women who have had an ovariectomy or a hysterectomy display a higher risk of PD. So, while estrogen may be protective for women, the higher rates in men may reflect a higher frequency of occupational toxicant exposures, e.g., farmers and horticulturalists, along with minor head traumas. Caffeine and tobacco partakers enjoy a measure of neuroprotection against the risk of PD [13,14,15].

### 1.2. Genetics

Although there are several genes known to be associated with PD, genetic information is not included in hospital discharge data. Thus, this dimension of PD epidemiology was not addressed.

### 1.3. Pesticides

Since the 1980s, a relationship between PD and exposure to various pesticides has been noted, both in general and in regard to specific pesticides. For insecticides, these included both organochlorine and arsenic insecticides, specifically chlorpyrifos, dieldrin, and rotenone, but not in relation to exposures with carbamates, pyrethroids, diazinon, malathion, or parathion. For herbicides, paraquat exposure has long been associated with PD, as has been trifluralin, 2,4-dichlorophenoxyacetic acid (2,4-D), 2,4,5-trichlorophenoxyacetic acid (2,4,5-TP), and nitrile and phenoxy herbicides, but not alachlor, atrazine, chlorophenylglycine, dicamba, glyphosate, phosphonoglycine, or triazine. For fungicides, increased risks have been noted with maneb and ziram, especially when they are deployed along with paraquat; no association has been found in relation to exposures with aromatic fungicides, dicarboximide, triazole, triflumizole, or vinclozolin. When it was possible to check for genetic susceptibility, it was found that any pesticide effect was increased in genetically susceptible individuals [16]. A meta-analysis of 19 case-control studies published between 1989 and 1999 reported a pooled risk of PD related to pesticides overall at 1.94 and without a dose–response relationship [14].

### 1.4. Pesticide Usage in Louisiana Agriculture

The major reference sources for insect control were the following editions of the Louisiana Insect Pest Management Guides: 1990, 1991, 1992, 1994, 1999, 2000, 2007, 2009, and 2017. For weed control, the major reference sources were the suggested chemical guides for 1988, 1989, 1991, 1992, 1994, 2000, 2002, 20003, 2007, 2009, and 2017. While the latter guides are used to both prevent and cull unwanted plants, insect control is primarily used retroactively, as and when an identified insect pest has exceeded a preset threat threshold density. Thus, reference to an insecticide indicates a possibility that it might have been deployed at that time while noting that the referenced insecticide was not the only insecticide recommended; repeated possibilities might approach past probabilities. Specific pesticide usage is discussed in Appendix A.

### 1.5. Other Exposures

It has been claimed that countries having paper, chemical, and iron- or copper-related industries have higher death rates for PD than countries without these industries. Long-term occupational exposure of more than 20 years to copper, manganese, or lead have been shown to be associated with PD. However, there is no convincing epidemiologic evidence that exposure to specific metals, e.g., copper, lead, iron, and manganese, cause PD sensu stricto [14]. In Mn-induced parkinsonism, the main target of neurotoxicity appears to be the globus pallidus rather than the nigrostriatal system, which is affected in idiopathic and familial PD cases [15]. Several studies have shown an increased risk of PD in individuals who suffered head trauma. The lag between the head trauma and the onset of PD was in the order of 21 years.

The principle objective of this research was to identify the present major pesticides in Louisiana impacting significant PD risk.

## 2. Materials and Methods

Clinical Records: The Louisiana Office of Public Health collects data from the Louisiana Hospital In-patient Discharge Database (LAHIDD) for the purpose of surveillance of diseases of public importance. Because of the public health surveillance needs, they are not required to have Institutional Review Board (IRB) approval for having access to surveillance-related databases. They process the files in-house behind the state firewall. The first task carried out by the surveillance staff is to eliminate duplicates of patients. Once de-duplication is completed, they remove personal identifiers to prepare their surveillance reports.

The patients used in this study were admitted to a Louisiana hospital between 1999 and 2012. There are about 500,000 hospitalizations per year in the 95 hospitals. Records were extracted for anyone with a Parkinson’s disease ICD10 (International Classification of Disease) code in any of the eight discharge diagnoses. These records were then ranked by order of admission date; the first admission was the only one used in the selection. Duplicate patient data were removed prior to de-identification. The Parkinson dataset provided to Dr. Hugh-Jones was completely de-identified, and all analyses were carried out on this de-identified file.

Limitations: Many PD patients are diagnosed and managed as outpatients and are therefore not in the hospital discharge diagnoses. Hospitalization is typically some 10 to 15 years after onset thanks to the complications of the advanced disease. Earlier hospitalizations will be from a possible coincident serious comorbidity, such as heart disease or cancer, and the PD diagnosis may or may not be noted among the discharge diagnoses. In the United States, women are proactive about seeking healthcare, and this could impact their early diagnoses. Thus, these hospital records are not necessarily a measure of the true extent of PD in Louisiana, but currently, it is all that is available.

Spatial Data: The only available spatial data in the patient records were the ZIP codes for residence; the street address was unavailable. It had to be assumed that the ZIP code, if a valid one, was the true ZIP code. False ZIP codes were compared to the nearest valid codes, and the error was usually obvious just based on common sense, e.g., switched order or misread typo. Patients whose ZIP code was for a post office box were assumed to live in the enclosing ZIP code area. ZIP code areas absent from the patient records were found on USPS.com at https://eddm.usps.com/eddm/customer/routeSearch.action. Census population numbers for 2000 and 2010 for each ZIP code area were obtained from ZIP-CODES.com at http://www.zip-codes.com/. The PD risk for each ZIP code area was calculated based on the number of patients diagnosed in the area in 1999–2005 and 2006–2012 divided by the relevant ZIP code population number in the 2000 or 2010 census and expressed as cases per 10,000. As will be discussed later, the 2006–2012 data reflect PD incidence and are of increased relevance compared with the historic “prevalence” of the 1999–2005 data.

In order to map the PD risk values by ZIP code area for each time period, a Microsoft Excel spreadsheet was joined with a copy of the ZIP code area polygons, using the ZIP code field as the foreign key. Once the records were joined, the PD risk values could be mapped using a graduated color map symbology (Figure 1 and Figure 2). Note: areas in these maps that seem to be missing data are regions without ZIP code areas.

In order to simulate a more realistic continuous representation of the PD risk values, interpolated predictive surfaces of risk values for PD were calculated. First, Geographic Information System (GIS) centroids (located within the polygon boundaries) were calculated for all the ZIP code polygons. The process employed for this task transferred the records for the polygons to the centroids. This step brought the PD values for each ZIP code area into the records of the centroids. Finally, interpolated predictive surfaces of PD risk values were calculated using the Empirical Bayesian Kriging (EBK) transformation, employing the K-Bessel detrended semivariogram model for both 1999–2005 and 2006–2012 datasets. For an explanation of Empirical Bayesian Kriging, see https://desktop.arcgis.com/en/arcmap/10.3/guide-books/extensions/geostatistical-analyst/what-is-empirical-bayesian-kriging-.htm.

To reduce the extended variance in the 2006–2012 EBK image due to denominator bias—i.e., very small ZIP code areas with zero to 500 population and with or without PD cases and thus calculated PD risk values at significant difference from larger immediate surrounding ZIP code areas—their data were merged with surrounding or adjoining areas, and their centroids were removed. This reduced the number of ZIP code centroids used from 529 to 450. Again, interpolated predictive surfaces of PD risk values were calculated using the EBK transformation and the edited sets of ZIP code centroids. The final PD risk interpolated surfaces appear in Figure 3 and Figure 4. All GIS tasks and geospatial analyses in this research investigation were performed within ArcGIS Advanced Desktop 10.3.1.

The EBK transformation employed in ArcGIS produces a model that, in addition to the interpolated surface, includes contours calculated from the ZIP code area GIS centroids. The predictive PD risk values that have been interpolated across the entire surface will vary slightly with the original values, at the exact locations of the source ZIP code centroids. The calculation of the contour locations is based on the exact values and locations of the centroids. This explains the difference in the ranges of values that appear in the keys of these two data layers (Figure 4 and Figure 5). To compare the EBK transformation model contours with contemporary agricultural crops, the contours were overlaid over a land cover image of the distribution of agricultural crops in Louisiana (Figure 5). This image was downloaded from the United States Department of Agriculture National Agricultural Statistics Service website https://nassgeodata.gmu.edu/CropScape/ and brought into ArcGIS for map production. The USDA National Agricultural Statistics Service (NASS) Cropland Data Layer (CDL) is provided to the public as is and is considered public domain and free for redistribution. USDA NASS does not warrant any conclusions drawn from these data. The metadata for each CDL state per year is online at https://www.nass.usda.gov/Research_and_Science/Cropland/metadata/meta.php. Each metadata record contains multiple statements about having no use constraints.

Water Quality Data. The drinking water in Louisiana is sampled and tested by the Louisiana Department of Health and Hospitals (LDHH); surface waters are sampled annually, and ground (well) waters are sampled every third year. Repeat sampling is done whenever a sample shows a pesticide detection at a level of half or higher of the federal maximum contaminant value allowed for the pesticide concerned. Quarterly sampling continues until the problem has been identified and corrected.

The levels of the following pesticides are measured: (**Bold** are known PD enhancers; normal script pesticides may be possible PD enhancers; and those in italic script are known not to induce PD.)
Fungicides:hexachlorobenzene (HCB), hexachlorocyclopentadiene. Herbicides:**2,4-D**, 2,4,5-TP, *atrazine*, dalapon, dinoseb, diquat, endothall, *glyphosate*, Lasso^®^, pentachlorophenol, picloram, simazine.Insecticides:**BHC-Gamma**, *carbofuran*, endrin, *oxamyl*, toxaphene.

The following pesticides were tested for, but no positives were found: chlordane, heptachlor, heptachlor epoxide, and methoxychlor.

Environmental Protection Agency (EPA) standard environmental chemistry methods (ECM) are used to identify and measure each of the above pesticides (https://www.epa.gov/pesticide-analytical-methods/environmental-chemistry-methods-ecm-index-0-9). The quality of this work is recertified each year by the EPA and checked for compliance (https://www.epa.gov/compliance/good-laboratory-practices-standards-compliance-monitoring-program). Because of the damage done by Hurricane Katrina in 2005 to the New Orleans Central Laboratory at 325 Loyola Ave., analyses were performed by certified laboratories in Arkansas and Texas after September 2005 until analytical work was moved to the recertified laboratory in Metairie, LA, in 2009. Since spring 2015, a certified contract laboratory has performed a large portion of the analyses, including those related to pesticides. The Louisiana Department of Health, Office of Public Health, Engineering Services generously provided a copy of the sampling and testing data for 8 June 1993 through 30 June 2016.

Estimates of Pesticide Use. As the actual use of pesticides is not recorded, estimates of the pesticide quantity data in the United States Geological Survey (USGS) Pesticide National Synthesis Project were used. These data are derived from aggregated proprietary farm survey pesticide-use data and are reported at the multi-county crop reporting district (CRD) level (https://pubs.usgs.gov/ds/752/). Harvested-crop acreage data by county from the U.S. Department of Agriculture Census of Agriculture are used to calculate the median pesticide-by-crop use rates for each crop in each CRD. These rates are applied to the harvested acreage of each crop in a county to obtain pesticide-use estimates at a county level. Two different estimation methods, EPest-low and EPest-high, are used to estimate a range of pesticide use. Both EPest-low and EPest-high methods incorporate proprietary surveyed rates for CRDs, but EPest-low and EPest-high estimates differ in how they treat situations when a CRD was surveyed and pesticide use was not reported for a particular crop present in the CRD. In these situations, EPest-low assumes zero use in the CRD for that pesticide-by-crop combination. EPest-high, however, treats the unreported use for that pesticide-by-crop combination in the CRD as missing data. In this case, pesticide-by-crop use rates from neighboring CRDs or CRDs within the same region are used to estimate the pesticide-by-crop EPest-high rate for the CRD.

## 3. Results

### 3.1. Hospital Records

The fourteen-year primary discharge data revealed 23,224 primary PD diagnoses for the period of 1999–2012, with 11,456 (49.3%) males and 11,768 (50.7%) females. Because the primary diagnosis was based on when an individual was first noted in the digital database for the period of 1999–2012, it did not record any earlier hospital discharges. It was therefore made of both old and new cases. The former cases were of uncertain onset, whereas the latter are clearly recent. To better appreciate this, the patient records were broken down by age and sex for each year of primary diagnosis (see Table 1). In four of the five initial years, 1999–2003, there were over 2000; the reason for the lower number, 1282 records, in 2001 is presently obscure. The larger number in 2002 logically included patients’ primary diagnoses missed in 2001. Thereafter, the annual numbers decreased to reach relatively stable numbers in 2006–2007, which are essentially level thereafter at least for the male patients. More simply put, for the first seven years, the patients included some 46% PD survivor patients who had returned to the hospital, but because of the lack of prior records in the 1999–2012 database, they were defined, falsely, as primary discharges. The normal annual incidence of true primary PD hospital discharges is in the region of 2.90 patients per 10,000 people in Louisiana, 3.03 for men and 2.79 for women.

When the records are then divided between 1999–2005 and 2006–2012, differences in the character of the patients in the two periods can be seen. When viewed by risk of PD per 10,000 in each ZIP code area (see Table 2), it is clear that, in the initial period, the patients had accumulated from prior years, but in the second period, which is now essentially counting just true new hospital cases, the risk-levels had shifted slightly to the left to lower, but probably more accurate, levels. The long tails to the right are a result of “denominator bias”, when the ZIP code area has a small population of only 100 to 500 individuals. When tabulated by age and sex, an interesting divergence emerges (see Table 3). During 1999–2005, 48% were male patients and 52% female, and in 2006–2012, 51% were males and 49% females. This seems to have arisen from the markedly better survival of female PD patients 80 years and older than of males. The latter had to survive to be diagnosed and were failing. For example, the modal age for men was 79 years and that for women was 80 years, but for those 10 years older, the male primary diagnoses were 32% of the modal value, whereas they were 45% for women. Overall, it was more commonly diagnosed in males 50 years and older until age takes its toll. For women, the age threshold was 52–53 years. Taking the 2006–2012 age range as a better indicator of true incidence, the numbers affected below the age of 50 years, i.e., familial cases, were very similar, 2.04% for males and 2.01% for female patients.

### 3.2. Spatial Distribution of Parkinson’s Disease

Figure 1 and Figure 2 show the distribution of PD diagnoses by ZIP code areas across the state of Louisiana for the two time periods, 1999–2005 and 2006–2012. In general, there is no obvious difference other than more diagnoses, as commented on earlier, except noticeably in a band between Lafayette and Monroe during 1999 to 2005. They were distinctly fewer in this band in 2006–2012. This is perceived more clearly when the EBK images are compared (see Figure 3 and Figure 4), but note that the color-coding scales differ to accentuate the different levels of risk. The spatial relationship between the 2006–2012 PD risk values and the distribution of agricultural crops in 2005 is shown in Figure 5; 2005 was chosen because of the delay between pesticide exposure and the onset of PD. The PD high-risk areas match closely the arbor-pastoral areas of the state that are of deciduous and evergreen forests, forest not otherwise specified, and grass/pastures.

In addition, the following was observed:The singular highest-incidence area in Figure 4 contains Mamou, Mittie, Oakdale, and Oberlin (Allen and Evangeline Parishes) and is part of a band from the east of the Atchafalaya River and in the west to DeRidder (Beauregard Parish), Starks (Calcasieu Parish), and the Sabine River. Within this high-risk area are numerous pastures and dense timber. In the east, it hooks south from Livonia to Rosedale, an area that also includes soybeans and sorghum.There is a second band of high risk from Sabine Parish, also on the Sabine River and the Texas border, which reaches northeast to West Carroll Parish and the Arkansas border and is similarly well populated with pastures and commercial woodlands. West Carroll has a higher PD risk level than Morehouse and East Carroll on either side. These latter parishes are characterized by agricultural crops, except in the far west of Morehouse with forestry, while West Carroll has forestry along its length but especially in the northern two-thirds, with cotton, soybeans, and sorghum in the bottom third; the water drains from north to south from similar woodlands in Chicot County, Arkansas. The 2006–2012 PD risk map also shows an area of increased PD risk south of Monroe in Caldwell Parish, at the time and place of intense cotton growing.In Figure 4, 2006–2012, it can be seen in the Florida parishes that there is an eastern cluster of high risk stretching down through Washington, northeastern St. Tammany, and Tangipahoa parishes to St. John the Baptist, northern St. Charles Parish, and northwest Jefferson Parish, west of New Orleans. As the ZIP code areas reach out into Lakes Pontchartrain and Maurepas, the calculated risks follow but are, in fact, two clusters. In the eastern Florida parishes, central Tangipahoa Parish is traditionally a strawberry growing area, but otherwise, it is pastures with woodland along with St. Tammany and Washington Parishes. Then, there is the “Western New Orleans” cluster of St. John the Baptist, northern St. Charles, and Kenner, where the first two parishes contain petrochemical plants and storage, shipbuilding, and some agriculture and Kenner is a dormitory suburb providing workers to the former parishes.There is a noticeable low-incidence “sink” to the west of Alexandria in northeast Vernon Parish, south of the Kisatchie National Forest, corresponding to Fort Polk and its non-resident young soldiers under training, as well as along the parishes west of the Mississippi River from East Carroll Parish in the northeast down to Concordia and then on the east side of West Feliciana; after a slight increase on the West and East Feliciana border, Iberville and St. Martin Parishes are essentially free of cases, as well as the lower Atchafalaya basin down to Houma. The relative absence along the length of the Mississippi is clear but with some risk in Jefferson (Kenner), St. John the Baptist, and (northern) St. Charles Parishes, as noted above. In 1999–2005 (Figure 3), the northeast parishes along the Mississippi were once afflicted with Parkinson’s disease patients but are now relatively free. This is an area where cotton and soybeans were traditionally grown. PD is restricted to the northern third of the rice growing areas, suggesting that any PD risk may be from a north–south surface flow of dangerous pesticides into the rice growing area and not from pesticides used on rice itself. PD is essentially absent from the sugarcane growing areas of the lower Mississippi from East Baton Rouge south, along the Lafourche River, and the western bank of the Atchafalaya basin from Breaux Bridge south.

For a detailed map of Louisiana showing towns and parishes (counties) and the Atchafalaya, go to http://ontheworldmap.com/usa/state/louisiana/large-detailed-map-of-louisiana-with-cities-and-towns.jpg.

While there may be a PD risk in relation to strawberries and the non-use of glyphosate-resistant crops, the major agricultural pesticide risk appears to be in relation to pesticides used in the arbor–pastoral areas, whether in the eastern Florida parishes, across central Louisiana, or from the Sabine river northeast to the northeast corner of Louisiana. On the other hand, rice and sugar cane crops appear to be PD-risk-free in Louisiana.

Utilizing the “Pulp & Paper Mill Map of SE”, it was possible to visualize the locations of the 11 mills in Louisiana (see https://www.google.com/maps/d/viewer?mid=1YDB1x49qf5cpKqQM0VU6DMZxdso&hl=en_US&ll=32.06594026455853%2C-86.772897&z=5).

Five were clearly in areas of high risk (Campti, Hodge, West Monroe, Bastrop, and Deridder) and six at marginal risk (Bogalusa, St. Francisville, Lockport, Zachary, Pineville, and Mansfield). Generally, there were no paper mills in areas of trivial risk, but this may reflect that such mills are placed where the source materials are located and not where the demand for the product is highest.

There are 19 sawmills in Louisiana (see AMFIBI Business Directory, http://www.amfibi.com/us/LA/10871-05ef043d/20. Only two of the 19 are in areas of low risk (both in Shreveport). The rest are in areas of modest- to high-risk, with nine north of the I-10 highway from Port Allen to Orange on the Texas border (Dequincy, Kinder (×2), Maringouin, Merryville, Morrow, Pitkin, Starks, Vinton), five north of Alexandria (Coushatta, Homer, Marthaville, Olla (×2)), two in Bogalusa, and one in Jefferson Parish (Metairie). There was no obvious clustering of PD cases.

### 3.3. Pesticides and Drinking Water

LDHH sampling of drinking water is not intense, being performed at a rate of once per year per surface water site and only every third year for groundwater sites, which is less useful, as any agricultural pesticide contamination is likely to be short (2 weeks). For the period from 1993 through 2012, 1775 surface water samples and 11,918 ground water samples were collected and tested. The pesticides tested for were the following: **2,4,5-TP**, **2,4-D** Atrazine, BHC-Gamma, Carbofuran, Dalapon, Dinoseb, Diquat, Endothall, Endrin, Glyphosate, Hexachlorobenzene, Lasso, Oxamyl, Pentachlorophenol, Picloram, Simazine, and Toxaphene (pesticides in **bold** are known to be associated with PD, those in italics are not associated). Only two (3% positive) out of 74 surface water samples in 2009 were found to contain 2,4-D herbicide; the levels noted were 0.5 µ/L for a St. Bernard Parish sampling site and 0.1 µ/L for a Sabine Parish site, collected on the same April day. 2,4,5-TP was found in five (0.8% of 601 samples tested) groundwater samples collected in 1996 and in two (0.2% of 976 water samples tested) groundwaters sampled and tested in 2010; in 1996, a cluster of five wells in Concordia was sampled on the same day in January and had identical 2,4,5-TP levels of 0.05 µ/L. The two 2010 positive samples were from wells in Vermilion Parish (0.527 µ/L) and Sabine Parish (0.285 µ/L), both sampled in April. Of the four parishes named, St. Bernard and Concordia were of negligible PD risk. There were too few positive samples to form a pattern, but the positive samples are proof of the concept that, in Louisiana, agricultural pesticides can and do find their way into drinking water.

### 3.4. Arbor–Pastoral Risk and Estimated Pesticide Use

A dozen parishes include the high-risk arbor–pastoral parts of Louisiana and are comprised of commercial forests and crops, woodlands, and grass/pastures. The USGS-estimated pesticide usages indicated that 2,4-D, chlorpyrifos, and paraquat, all known PD-associated pesticides, were dispersed in these parishes (see Table 4 for four example years, 1992, 1996, 2000, and 2004). 2,4-D was present in the largest amounts in each parish, whether using the low or the high estimates for the year and thus was of the highest certainty of eventual human exposure. The dispersal across any parish would not have been even but would, in fact, have been in scattered, defined areas where the pesticide was actually needed at the time or times it was used.

### 3.5. Potential Aquifer Recharge

It is interesting that these arbor–pastoral PD high-risk areas of Louisiana also correspond to high and moderate aquifer recharge potential (see Figure 6). This map shows the aquifer recharge potential across the state of Louisiana. The potential risks of aquifer contamination in recharge areas are significant. Aquifer recharge occurs primarily by the direct infiltration of precipitation in the outcrop–subcrop areas of the geologic formations that constitute the aquifer system. They also receive recharge by leakage from surface water bodies in outcrop areas, the downward percolation of water through overlying surface materials in subcrop areas, and water movement from streams into aquifers when stream stages are above aquifer water levels, all of which describe how these arbor–pastoral pesticides would get into the recharge waters to be consumed by the resident human population.

## 4. Discussion

This study was based on some 23,224 hospital-discharge PD diagnoses of 23,224 individuals from Louisiana hospitals for 1999 through 2012. The early years, by the nature of the database, included individuals who were recorded on a revisit as well as those truly on their first visit. By 2006 the revisitors had been all but screened out, and for 2007–2012, we had a fair view of the true Louisiana annual incidence of PD in the region of 2.90 patients per 10,000 people, 3.03 for men and 2.79 for women. These are slight underestimates, as they are based on patients seen in hospitals with a recorded diagnosis of PD. Missing will be those individuals with PD, but whose condition was not diagnosed, as well as Louisiana residents who had been diagnosed elsewhere with PD and whose condition had yet to be recorded by a Louisiana hospital. The percentage of PD cases from a familial genetic cause is very similar for both sexes, 2.04% for men and 2.01% for women patients, using the 2006–2012 data for those under 50 years of age. The close similarity of the two values is very persuasive for a genetic cause or predisposition.

## 5. Conclusions

The major risks identified came from the pesticides used in relation to areas of forestry, woodlands, and pastures and were from 2,4-D and from paraquat and chlorpyrifos. These extensive rural areas overlay the areas of significant potential aquifer recharge within Louisiana, which would have ensured human exposure whenever the pesticides were deployed. This emphasizes the need for rigorous oversight of the present and future pesticides permitted.

Presently, it appears that PD is no longer associated with growing cotton and soybeans, thanks to the introduction of glyphosate- and Roundup^®^-resistant cotton, soybeans, and corn, a significant benefit. The prior cause may have been the pre-emergent use of trifluralin; paraquat and 2,4-D may have also played a part. In addition, the relative values of these crops vary from year to year; cotton is no longer as profitable as it once was and has been largely replaced by soybeans and sorghum. With these fluctuations, the selection of pesticides will vary and with them the risk (see the cluster south of Monroe and that around Livonia west of Baton Rouge). Herbicides appear to continue to be associated with PD in Louisiana in relation to their use in commercial timber sites, woodlands, and pasture weed control. This is clearly of major concern because of the extensive areas involved throughout the state and the unending crop management demands, thus resulting in persistent risk. The risk associated with strawberries is unclear and may just reflect the drainage of pesticides across the strawberry farms from surrounding areas.

At this point, the consistent association of PD risk with arbor–pastoral areas strongly suggests an association with the pesticides used on trees and pastures. The use of the insecticide chlorpyrifos, once widely advised for pine trees, has been restricted since 2000, and the USGS data show that it is disappearing. The use of paraquat, a herbicide with long-recognized PD-risk, would appear from the USGS estimates to be equally modest. While imazapyr and imazamox may be of concern because of their frequent use, presently, neither is listed as a PD-inducing pesticide, and imazapyr-resistant rice is a preferred crop. However, 2,4-D is deployed as a tree injector and can be used year round, though spring and summer is preferred, and throughout the year for weed control. The USGS’ low estimates of its use in the arbor–pastoral parishes show it in metric tons in 9 of 12 parishes for the four example years in Table 4. Logically, any future study on this matter should also investigate the contribution, if any, of the Sabine River and Texas agriculture, as the Sabine River directly abuts high-risk Louisiana. One does not have to be living on a riverbank to be drinking its water. There are a number of Native American communities in that area and at risk.

The affected St. John the Baptist and St. Charles Parishes and Kenner in western Jefferson Parish are a mixture of small towns, industrial petrochemical plants, agriculture, and suburban sprawl, so the possible cause of their elevated risk is not immediately obvious. As the upstream parishes adjoining the Mississippi are essentially free of PD, it is logically a cause singular to these noted downstream parishes.

The persistence of a very low-risk area to the west of Alexandria suggests that the local beneficial causes are constant and important. This comprises Fort Polk, a military training camp with a largely transient population of young soldiers, and if they were exposed, any emergence of PD would be years later and many miles away. There is also a band of low risk to the north and east of Alexandria. The absence of PD in relation to the major rice and sugar cane areas suggests a minimal use of PD-enhancing pesticides in these areas. This disruption in the PD distribution indicates that lessons learned here as to the causes of low or minimal risk might be applicable elsewhere throughout the state.

The essential minimal levels of PD along significant stretches of the Mississippi River, and in Morgan City and Houma, show that the river’s water is safe regarding PD risk.

The results of this study raise several questions and relevant future objectives:The need to confirm the new apparent freedom from PD in the cotton, rice, sugar cane, and soybean areas in certain parishes. Why does there appear to be a persistent high risk in discrete other places in relation to cotton, rice, and soybeans and possibly other crops?The need to identify and quantify the risks associated with the high-incidence band east to west from the upper Atchafalaya to DeRidder and Starks on the Sabine River. The risks appear to be constant over time.The need to identify the herbicides most probably associated with the elevated PD risks in St. Tammany, Tangipahoa, and Washington Parishes, as well as upstream to the north in adjoining Mississippi counties and in the other arbor–pastoral parishes across Louisiana from Sabine to West Carroll.The need to identify and quantify the cause or causes of the relative freedom from PD in that discrete area in Vernon Parish to the north of Fort Polk and to the west of Alexandria and comparatively to the north of the latter city.The need to identify the local cause or causes of the higher risk of PD in Jefferson, St. John the Baptist, and St. Charles Parishes, which are possibly of an industrial nature.The need to share this information, where relevant, with the appropriate state departments in Arkansas, Mississippi, and Texas.

It should be kept in mind that, in many ways, this is a retrospective study of pesticide risks for Parkinson’s disease regarding chemicals deployed in past decades—in the 1980s and 1990s primarily—and not necessarily today. However, today, we still carry their costs in human cases. Present pesticides should not add to our pain.

There are three aspects of Parkinson’s disease that need to be kept in mind: (1) It is, unfortunately, a disease that cannot be cured; it can only be prevented. (2) It is part of the agricultural ecology. (3) It is dynamic; its expression can change. For example, in Serbia, in 2010, it was positively associated with exposure to insecticides, with gardening, and with drinking well water and spring water [17]. In California in 2009, it was shown to be associated with consuming water from private wells in areas with documented historical pesticide use, with a 70%–90% increase in relative risk of Parkinson’s disease [18]. This was not a unique observation and was in relation to agricultural crops. It is not unlikely that well water was the cause of pesticide exposure in Louisiana. In France, vineyards rank among the crops that require most intense pesticide use, and regions with greater presence of vineyards are characterized with higher Parkinson’s disease risk [19]. In Israel, the standardized incidence rates (SIR) for Parkinson’s disease were higher than expected in Jewish rural areas, with the highest SIRs in proximity to agriculture cultivated fields and to field size [20]. In Louisiana, we have witnessed a major reduction in Parkinson’s disease incidence and distribution thanks to the introduction of glyphosate-resistant crops. Now, we need to firmly limit the use of 2,4-D, paraquat, and chlorpyrifos to prevent further cases.

## Figures and Tables

**Figure 1 ijerph-17-01584-f001:**
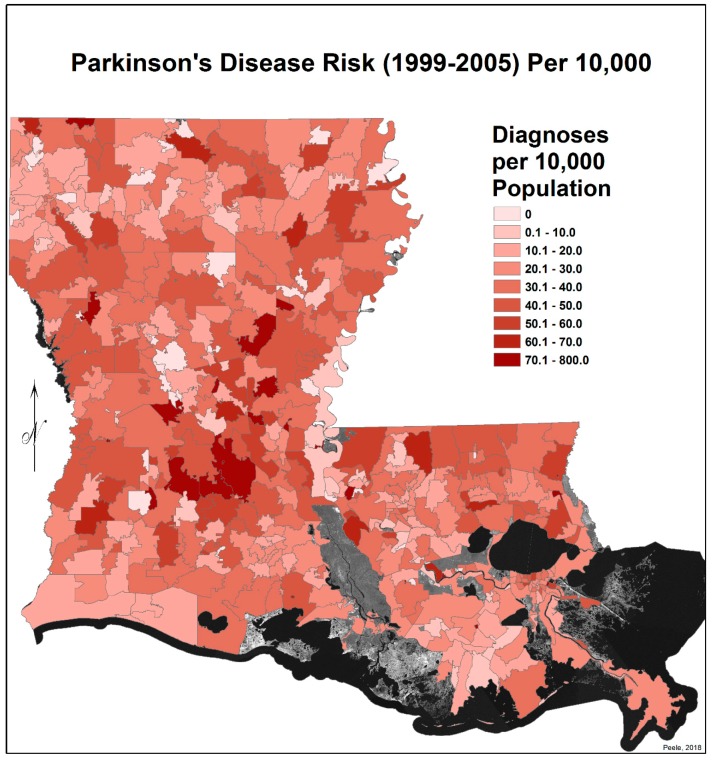
Parkinson’s disease risk for 1999–2005 per 10,000 persons by Louisiana ZIP Code.

**Figure 2 ijerph-17-01584-f002:**
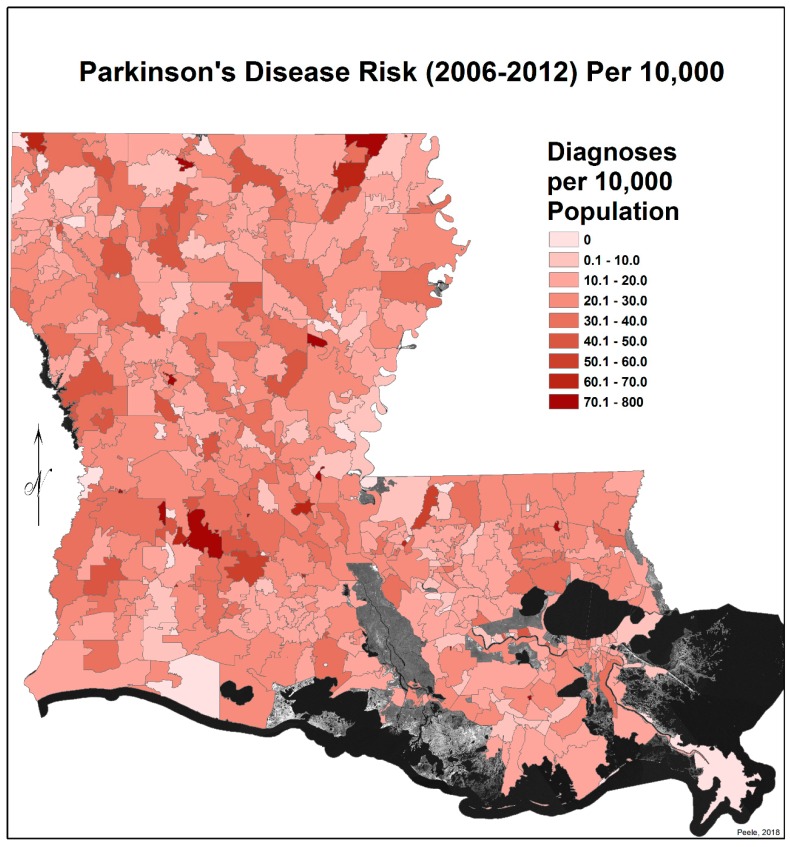
Parkinson’s disease risk for 2006–2012 per 10,000 persons by Louisiana ZIP Code.

**Figure 3 ijerph-17-01584-f003:**
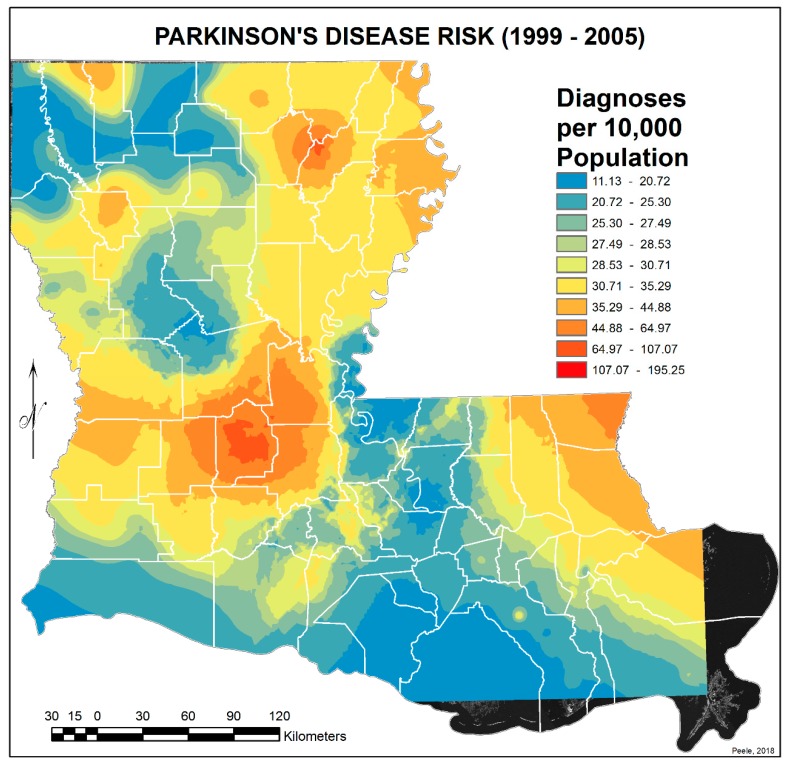
Empirical Bayesian Kriging interpolated predictive surface of Parkinson’s disease (PD) risk values derived from the ZIP code-based Parkinson’s disease risk data shown in Figure 1 for 1999–2005.

**Figure 4 ijerph-17-01584-f004:**
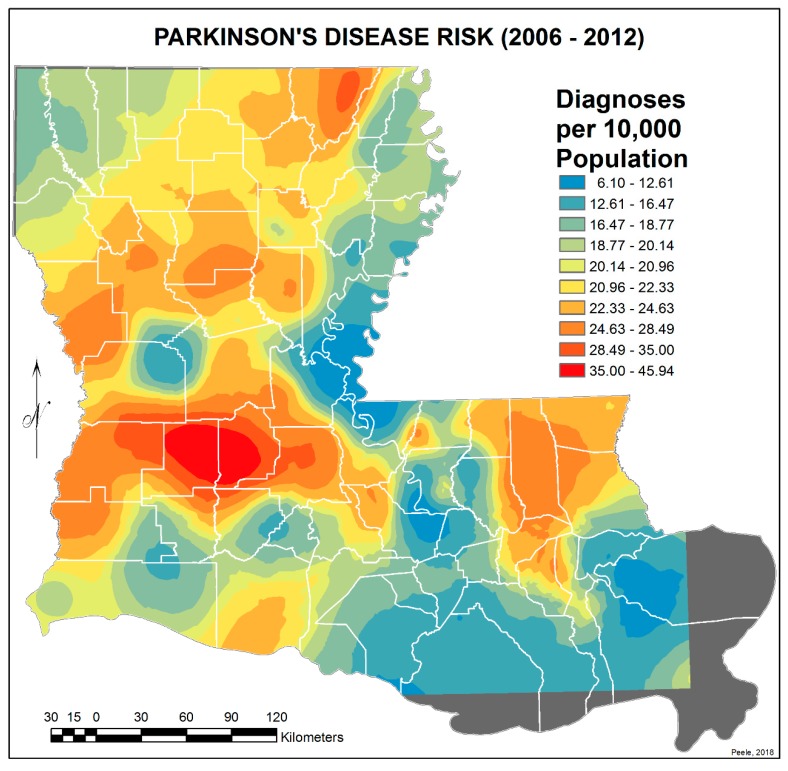
Empirical Bayesian Kriging interpolated predictive surface of PD risk values derived from the ZIP code-based Parkinson’s disease risk data shown in Figure 2 for 2006–2012 but adjusted to account for small areas with zero or limited populations.

**Figure 5 ijerph-17-01584-f005:**
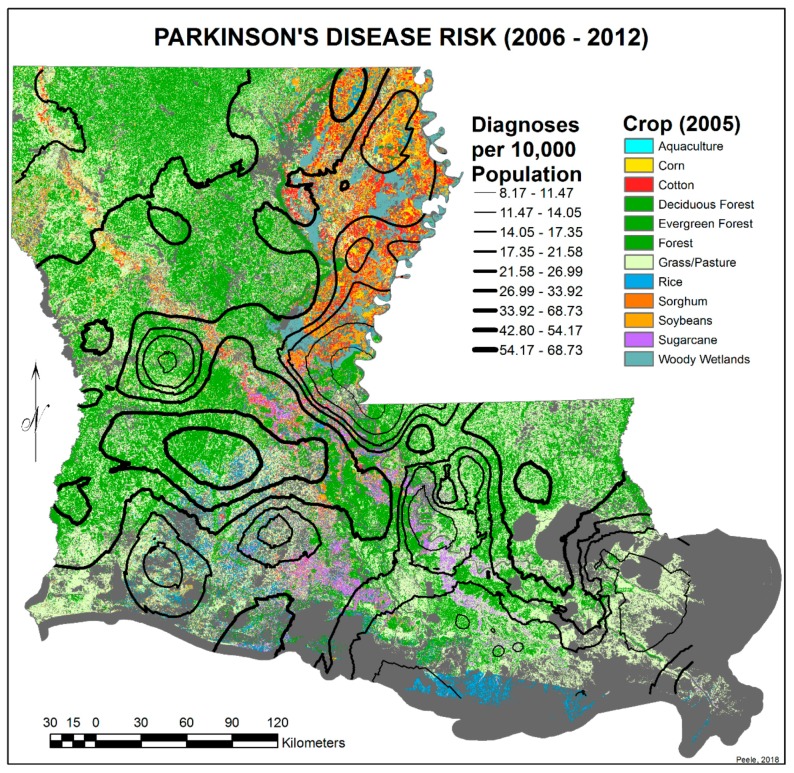
Risk of Parkinson’s disease in Louisiana in 2006–2012 in relation to the agricultural crops of 2005.

**Figure 6 ijerph-17-01584-f006:**
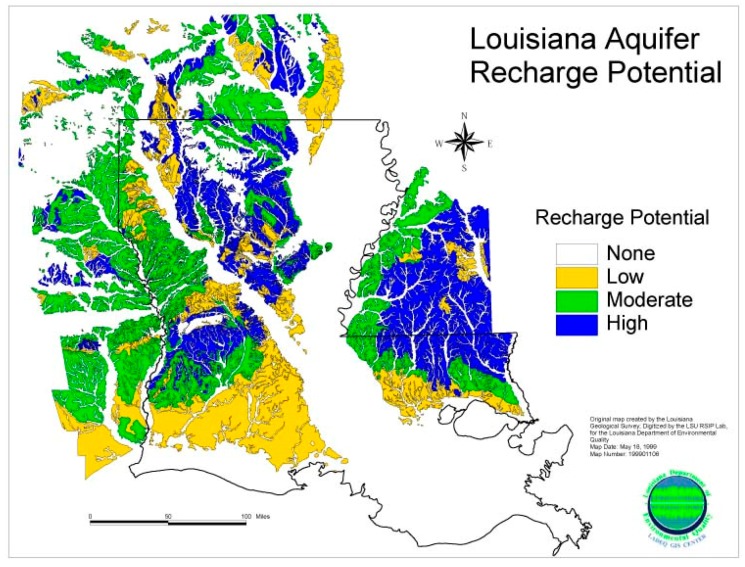
Louisiana aquifer recharge potential. Source: Water: Recharge Potential of Louisiana Aquifers Maps, Louisiana Department of Environmental Quality. 1989. https://deq.louisiana.gov/page/recharge-potential-of-louisiana-aquifers-maps.

**Table 1 ijerph-17-01584-t001:** Louisiana hospital primary discharge diagnoses of Parkinson’s disease, 1999–2012.

					Years									
	1999	2000	2001	2002	2003	2004	2005	2006	2007	2008	2009	2010	2011	2012
Primary PD Discharges														
All	2427	2087	1282	2.480	2106	1940	1582	1469	1270	1306	1349	1289	1332	1305
Male	1179	952	668	1154	1010	937	804	747	630	641	674	679	690	691
Female	1248	1135	614	1326	1096	1003	778	722	640	665	675	610	642	614
Estimated Population in Millions														
Total	4.461	4.472	4.478	4.497	4.521	4.552	4.577	4.303	4.376	4.436	4.492	4.546	4.576	4.605
Male	2.147	2.165	2.170	2.182	2.194	2.213	2.227	2.099	2.136	2.166	2.196	2.226	2.238	2.252
Female	2.314	2.310	2.308	2.316	2.327	2.339	2.349	2.204	2.239	2.270	2.295	2.320	2.338	2.353
Results/10,000														
All	5.44	4.67	2.86	5.51	4.66	4.26	3.46	3.41	2.90	2.94	3.00	2.84	2.91	2.83
Male	5.49	4.40	3.08	5.29	4.60	4.29	3.61	3.56	2.95	2.96	3.07	3.05	3.08	3.07
Female	5.39	4.92	2.66	5.73	4.71	4.23	3.31	3.28	2.86	2.93	2.94	2.63	2.75	2.61

Sources: Louisiana estimates, 1999: Population estimates for the U.S., regions, and states by selected age groups and sex: annual time series, 1 July 1990 to 1 July 1999 (includes revised 1 April 1990 population counts) http://www.census.gov/popest/data/state/asrh/1990s/tables/ST-99-09.txt; Louisiana estimates, 2000–2010: Intercensal estimates of the resident population by sex and age for Louisiana, 1 April 2000 to 1 July 2010 (ST-EST001NT-02-22). Source: U.S. Census Bureau, Population Division. Release Date: October 2012. http://www.census.gov/popest/data/intercensal/state/ST-EST00INT-02.html; Louisiana estimates, 2010–2012: American Fact Finder, U.S. Department of Commerce, United States Census. http://factfinder.census.gov/faces/tableservices/jsf/pages/productview.xhtml?src=bkmk.

**Table 2 ijerph-17-01584-t002:** Parkinson’s disease risks by Louisiana ZIP code areas.

Risk per 10,000	ZIPs 1 (1999–2005)	ZIPs 2 (2006–2012)
0–9	39	43
10–19	32	46
20–29	86	167
30–39	111	133
40–49	102	63
50–59	65	26
60–60	32	2
70–70	12	7
80–89	2	7
90–99	9	3
100–149	3	2
150–199	10	4
200–249	2	1
250–299	0	1
≥300	1	2

**Table 3 ijerph-17-01584-t003:** Louisiana hospital primary PD discharges by age and sex.

	1999–2005			2006–2012		
Age in Years	Male	Female	Total	Male	Female	Total
<20	3	1	4	2	0	2
20–29	11	6	17	11	10	21
30–39	30	19	49	17	16	33
40–49	74	64	138	67	66	133
50–59	318	219	537	342	257	599
60–60	945	832	1777	860	685	1545
70–79	2751	2617	5368	1717	1511	3228
80–89	2248	2828	5076	1532	1696	3228
90–99	321	600	921	203	322	525
≥100	3	14	17	1	5	6
Totals	6704	7200	13,904	4752	4568	9320

**Table 4 ijerph-17-01584-t004:** USGS-estimated annual agricultural pesticide use for arbor–pastoral parishes (counties) in Louisiana, 1992–2004, from https://pubs.usgs.gov/ds/752/ For parish locations and relation to vegetation, see https://geology.com/county-map/louisiana.shtml.

(a) 2,4-D								
Louisiana Parish	2,4-D Kg, Epest-low	2,4-D Kg Epest-high
1992	1996	2000	2004	1992	1996	2000	2004
Allen	6535.5	4426.4		4050.4	7175	5781.5		4209.9
Beauregard	966.8	1876	4260.2	2441.5	2454.3	4281.3	4390.6	2683.3
Bienville	369.1		3217.2	6535.5	622.5	465.1	3234.3	377.5
Claiborne	12.5		4679	0.6	316	596.1	4679.2	552.3
East Feliciana	792.8	809.2	360.9	700.1	1503.4	1006.3	938.4	734.4
Grant	221.8	374.2	160.4	768.1	383.9	942	200.4	922.2
La Salle	745.3	213.8	32.8	158.5	785.1	301.5	32.8	158.5
Sabine	165.7	26.1	1329.1	3602	464.2	779.8	1329.1	3602
St. Helena	2355.6	896.8	18	239.8	2553.2	1046.1	235.5	274.7
Tangipahoa	3926.9	1622.2	481.7	677	4507.3	1938.9	1035	734
Washington	2728.3	1407.7	353.6	433.1	3606.7	1718.6	688.8	508.6
Winn	3.8		1919.9		104.1	199.7	1919.9	221.9
**(b) Chlorpyrifos**								
**Louisiana Parish**	**Chlorpyrifos Kg, Epest-low**	**Chlorpyrifos Kg Epest-high**
**1992**	**1996**	**2000**	**2004**	**1992**	**1996**	**2000**	**2004**
Allen	24.9	10.3	35.2	2.5	67.6	151.3	58.7	13.8
Beauregard	70.7	46.1	313.5	44.7	187.1	637	443.7	89
Bienville	1.6	104.4		7.7	160.3	106.9	8	10.2
Claiborne		2.1		0.9	16.1	2.1	1.9	0.9
East Feliciana	13.9	486.6			182.8	502	141.4	112.2
Grant	479.3	504.3	416	22.8	491.5	1040.1	569.6	160.6
La Salle	7.1	3.7	3.2	4.4	15.6	4.4	5	7.1
Sabine	0.8	15		2.8	3.5	17.5	1.1	2.8
St. Helena	7.3	987.3	0.8		96.2	993.7	73.6	98.9
Tangipahoa	44.5	1441.8	0.6		283.1	1461.3	223.7	174.2
Washington	74.1	2238.8	3.4		449.9	2254.5	194.6	214.3
Winn	0.6	37.1		1.6	4.8	37.1		16
**(c) Paraquat**								
**Louisiana Parish**	**Paraquat Kg, Epest-low**	**Paraquat Kg Epest-high**
**1992**	**1996**	**2000**	**2004**	**1992**	**1996**	**2000**	**2004**
Allen				6.4	59.1	223.3	146.4	54.5
Beauregard	2.5	4.8	15.5	5.6	89.2	272.7	318.5	135.1
Bienville		4.2	1.8	69.6	449.8	27.1	3.3	69.6
Claiborne					711.8	14	0.6	
East Feliciana	259.1	45.1	20.1	5.4	277.9	95.9	165.3	112.8
Grant	50	86.3	699.7	45.9	160.1	117.7	793.6	174.6
La Salle	4.9	12.1	34.9	10.8	11.8	12.3	34.9	23.9
Sabine		0.5	1.9		1.7	17.1	6.2	
St. Helena	5.4	0.2	1.1	1.1	21	72.7	153.9	110.7
Tangipahoa	407.3	576.2	15.2	1.2	435.3	692.5	352	179.7
Washington	191.5	83.2	22.9	7.3	224.1	251.8	421.7	244.2
Winn					208.7	4.1

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
