# Peer review of "Parkinson’s Disease in Louisiana, 1999–2012: Based on Hospital Primary Discharge Diagnoses, Incidence, and Risk in Relation to Local Agricultural Crops, Pesticides, and Aquifer Recharge"

_ijerph, 2020, doi:10.3390/ijerph17051584_

Round 1
Reviewer 1 Report
I reviewed with Interest the extensive study looking at a correlation between Parkinson's disease prevalence and exposure to pesticide and herbicide. A similar study has previously looked at this correlation in the state of Nebraska but is likely the first study in State of Louisiana and is much more thorough and detailed providing zip code level analysis (compared to county level analysis for NE study). Advantage of the NE study was presence of a state mandated registry however the authors here have cleverly compensated for this lack through use of hospital discharge data. Although hospital discharge data for PD has been looked at previously (such as again in NE) but not for use as a corollary of PD prevalence. Authors describe in good details their method of ‘normalizing’ the data by looking at distribution on county borders and developing a ‘heat’ map with hot zone.
The paper also appropriately reviews available evidence on correlation (or lack of) with various pesticides and herbicides. Amount of data used is large. Limitation of using hospital discharge diagnosis for obtaining PD cases is properly identified.
Peticide usage is reasonably obtained from Pest Management guide but use of an agriculture registry would have been ideal as previously done in NE study.
Maps are very well designed and though out. The amount of data depicted clearly through the map is wonderful and refreshing to see.
Looking at the spread of diagnosed cases per year divided by male and female sex is the authors do a good job of explaining that the new cases diagnosed per year are higher initially given that there is no prior data available before 1999, however over the years as the patient's keep coming back and only the new patients are looked at the numbers drop of new cases admitted to the hospital and discharged with a diagnosis of Parkinson's disease. The authors test reasonably distinguish between cases before 2005 and after 2005 given the observed drop in blood year new cases around that time. It still however a week way of trying to estimate new incidence of Parkinson's versus a prevalence of Parkinson's although best that can be done and given data set.
It is interesting to note that male female distribution remains pretty consistent equal throughout the data with occasional predisposition to male he cases versus female such as in 2012 and authors make a good attempt in trying to explain it based on longer survival of the men's compared to men which will biased the hospital dissociate I given the admissions are more common in later age especially in the last few years of life. I think the authors to just focus on suggesting that men and women with Parkinson's disease have equal risk of being hospitalized for various health reason despite man having high risk of getting Parkinson's as the data from age below 50 shows that.
Table 2 needs some footnote to describe the table a little better. Especially when someone is directly looking at the tables.
Although there is an attempt by the authors to try to look at chronic distributions few years before the hospital discharges of Parkinson's patient to account for a delay in the onset of Parkinson's disease but there are many problems with that kind of analysis and this given data entry point out only one such problem the hospital admissions for patients with Parkinson's disease are significantly less for early Parkinson's disease then later Parkinson's disease and it is quite likely that a lot of patients will being admitted to the hospital have had Parkinson's for a while. There is no clear data in this data set on duration of disease of these patients were being discharged from Parkinson's but based on my personal experience and from some earlier published data the patient's 1 of the hospital for various reasons typically have disease duration for 5 to 7 years or longer which means that even looking at a data set from 2005 still is a very poor predictor of a risk of Parkinson's in 2012 given that the disease starts many years before the diagnosis and the patient is being admitted to the hospital already have this diagnosis likely for a long time. So any such sequential analysis ideally should not even be attempted with this data set but at minimum should be extremely cautiously discussed and not highlighted.
In the discussion section I would again suggest other precaution in trying to overemphasize that the data set is of true measure of annual incidence of Parkinson's disease in the region given that it is mostly dependent on hospital discharge. Many conditions that commonly result in hospital discharge is not actually less often seen in patients with Parkinson's disease lowering the need for hospitalization presumptively such as heart disease and cancers. Parkinson's disease itself without presence of any comorbidity is typically does not require hospitalization until very late in stage typically 10 to 15 years after the onset due to complications of advanced Parkinson's disease. It is quite common that the hospitalizations when the Parkinson's disease is put on the discharge diagnosis is still unrelated to PD and is mostly due to other comorbidities. Earlier work has reported on the common comorbidities that resulted in hospitalization patient Parkinson's disease in Nebraska when comparing hospital discharge state in Nebraska with the Nebraska state registry for Parkinson's. This is also supported by the fact that the difference between men and women found in the study is much lower than a fully established that men have almost 2 times high risk of developing Parkinson's disease in women in any ethnicity any race in any country. There presumption about familial genetic causes of patients under 50 years of age seems reasonable although it is pretty common for patients in their 40s to have idiopathic nonfamilial Parkinson's disease which is still much more common than genetic causes. The genetic causes typically strongly outweigh any idiopathic cause as an etiology below age 30. In this way of estimating genetic proportions of Parkinson's disease a very long shot and should not be over emphasized.
Author Response
Thank you very much for the quality and detail of your comments. It is greatly appreciated. Being a veterinarian with little experience of human diseases your explanation as to how and why PD cases acquire hospital diagnoses was revealing. Using you comments I have redrafted the Limitations paragraph, lines 132-135, in the Material & Methods
Limitations: Many PD patients are diagnosed and managed as outpatients and therefore not in the hospital discharge diagnoses. Hospitalization is typically some 10 to 15 years after onset thanks to complications of advanced disease. Earlier hospitalizations will be from a possible coincident serious comorbility, such as heart disease or cancer, and the PD diagnosis may or may not be noted among the discharge diagnoses. In the United States, women are proactive about seeking health care and this could impact their early diagnoses. Thus, these hospital records are not necessarily a measure of the true extent of PD in Louisiana, but currently it is all that is available.
The reference to American women proactively seeking medical aid more often than men, thereby increasing the possibility of a higher recording and certainly earlier recording of a Parkinson’s diagnosis, came from a recent New York Times article – Roni Caryn Rabin. Scientists Study Why The Illness Seems To Be Hitting Men Harder Than Women, February 21, 2020, Page A7. https://www.nytimes.com/2020/02/20/health/coronavirus-men-women.html?searchResultPosition=2
I doubt that it needs referencing as it is but a side comment.
Table 2, line 293: To make it more obvious what is being counted, I have changed Period to ZIPs. ZIPs 1 and ZIPs 2 reinforce that they are different periods without having to spell it out.
Reviewer 2 Report
Review Parkinson’s disease in Louisiana… IJERPH Hugh-Jones ME & al.
Main remark:
- Some data are difficult to follow for a reader outside Louisiana:
- Example 1 : lines 308-350: where are all these places situated on the map? (Mamou, Mittie, Oakdale,…, Atchafalaya basin). Impossible to situate if you are not familiar with Louisiana geography. It is worthwhile to find another way to situate these observations on the map. Maybe arrows can be added to the map to represent these observations.
- Example 2 : parishes: difficult to situate from ZIP code. Why do you use Parishes for Table 5? Maybe you can refer to a link to the map of parishes in Louisiana (as you have done for the mill map line 358-9).
- Ecological analysis: for an epidemiologist, this study is typically an ecological study. This type of study should be mentioned in the abstract and in the discussion.
- What does the black colour represent on maps 1 & 2?
Other minor remarks:
Line 57: Unfortunately, with modern image technology, PD tended to be overdiagnosed
Line 166: empirical Bayesian kring: a reference citation should be added
Table 4: redundant: sufficient to mention that, on a total of N Analysis from 1993 to 2010, 2,4, 5 TP was detected in 1996 and 2010 (n positive samples of N tested samples) and 2, 4 D in 2009 (n positive samples of N tested samples).
Lines 285-286: M > F 50 years and older until ages takes its toll after 80 yrs.
A question:
Line 127: 500 000 hospitalizations per year: how many hospitals in Louisiana?
Author Response
I much appreciate the quality of the reviewer’s comments and their advisory note. Such a pleasant and welcome change!
Review Parkinson’s disease in Louisiana… IJERPH Hugh-Jones ME & al.
Main remark:
- Some data are difficult to follow for a reader outside Louisiana:
- Example 1 : lines 308-350: where are all these places situated on the map? (Mamou, Mittie, Oakdale,…, Atchafalaya basin). Impossible to situate if you are not familiar with Louisiana geography. It is worthwhile to find another way to situate these observations on the map. Maybe arrows can be added to the map to represent these observations. There is a lot of detail but to get the message across the figures must be kept relatively simple and not cluttered with symbols and letters. In lines 357 – 359, I have provided a link to a jpg map of the state showing towns, parishes (counties), and the Atchafalaya, adequate for someone needing to know the locations mentioned.
- Example 2 : parishes: difficult to situate from ZIP code. Why do you use Parishes for Table 5? Maybe you can refer to a link to the map of parishes in Louisiana (as you have done for the mill map line 358-9). I have modified Table 4 (was 5) to provide a link to a parish map giving the names and in relation to the vegetation, which gives an added location aid, see lines 409 – 410. To find the vegetation plus parish outline map scan down the page.
- Ecological analysis: for an epidemiologist, this study is typically an ecological study. This type of study should be mentioned in the abstract and in the discussion.
‘Agricultural ecology’ has been added to the list of keywords (line 28), and is referred to in the final discussion paragraph (line 514).For me, epidemiology is merely the medical & veterinary version of ecology. It just has a different ultimate purpose, the better control of diseases. The purpose of this paper is to improve the control of PD by better awareness of the problem pesticides in Louisiana. - What does the black colour represent on maps 1 & 2?
- Essentially the Gulf of Mexico and coastal lakes.
Other minor remarks:
MR1 Line 57: Unfortunately, with modern image technology, PD tended to be over-diagnosed.
Noted and phrase inserted (line 57).
MR2 Line 166: empirical Bayesian kring: a reference citation should be added
Lines 170-172: For an explanation of empirical Bayesian Kriging see https://desktop.arcgis.com/en/arcmap/10.3/guide-books/extensions/geostatistical-analyst/what-is-empirical-bayesian-kriging-.htm
MR3 Table 4: redundant: sufficient to mention that, on a total of N Analysis from 1993 to 2010, 2,4, 5 TP was detected in 1996 and 2010 (n positive samples of N tested samples) and 2, 4 D in 2009 (n positive samples of N tested samples).
Table 4 was removed, as advised, along with its explanatory note about individual pesticide risks. A sentence was modified to now give the total surface water and ground water samples collected and tested; see lines 382-387. Table 5 is now Table 4.
MR4 Lines 285-286: M > F 50 years and older until ages takes its toll after 80 yrs.
Noted.
A question:
Q Line 127: 500 000 hospitalizations per year: how many hospitals in Louisiana?
95 hospitals are listed in Louisiana, see https://en.wikipedia.org/wiki/List_of_hospitals_in_Louisiana
I don’t think it needs to be referenced.
Reviewer 3 Report
The manuscript of Hugh-Jones and coworkers reports the findings supporting the role of environmental factors in the occurrence of PD. In particular, the authors by using information from patients admitted to a Louisiana hospital between 1999 and 2012 try to identify a correlation between pesticides and arbor-pastoral areas in Louisiana impacting significant PD risk.
Interestingly, authors show also that these arbor-pastoral PD high-risk areas identified in Louisiana correspond to high and moderate aquifer recharge potential, and then the potential risk of aquifer contamination has been also discussed.
I believe that the manuscript contains a clear map relating the PD risk and the various environmental factors present in the area in question, although I think that the study would have had a greater impact if the patients' genetic information had also been included.
Finally, I suggest that authors comment and discuss the data presented in other papers, in order to compare the results of this analysis with those known in other contexts.
Here are just a few examples of useful references.
Kab S, Spinosi J, Chaperon L, Dugravot A, Singh-Manoux A, Moisan F, Elbaz A.
Agricultural activities and the incidence of Parkinson's disease in the general
French population. Eur J Epidemiol. 2017 Mar;32(3):203-216.
Gatto NM, Cockburn M, Bronstein J, Manthripragada AD, Ritz B. Well-water consumption and Parkinson’s disease in rural California. Environ Health Perspect. 2009;117(12):1912–8.
Yitshak Sade M, Zlotnik Y, Kloog I, Novack V, Peretz C, Ifergane G. Parkinson’s disease prevalence and proximity to agricultural cultivated fields. Parkinsons Dis. 2015;2015:576564.
Vlajinac HD, Sipetic SB, Maksimovic JM, Marinkovic JM, Dzoljic ED, Ratkov IS, Kostic VS. Environmental factors and Parkinson's disease: a case-control study in Belgrade, Serbia. Int J Neurosci. 2010 May;120(5):361-7. doi: 10.3109/00207451003668374.
Ball N, Teo WP, Chandra S, Chapman J. Parkinson's Disease and the Environment.
Front Neurol. 2019 Mar 19;10:218.
Angelopoulou E, Bozi M, Simitsi AM, Koros C, Antonelou R, Papagiannakis N,
Maniati M, Poula D, Stamelou M, Vassilatis DK, Michalopoulos I, Geronikolou S,
Scarmeas N, Stefanis L. The relationship between environmental factors and
different Parkinson's disease subtypes in Greece: Data analysis of the Hellenic
Biobank of Parkinson's disease. Parkinsonism Relat Disord. 2019 Oct;67:105-112.
Author Response
Your advice is much appreciated. I have followed your advice and added a terminal paragraph using four of your suggested references. At this time I tried to add this paragraph to your system’s page but it did not allow it. The present line numbers for this new paragraph are 513-527.
There are three aspects of Parkinson’s disease that need to be kept in mind: It is, unfortunately, a disease that cannot be cured, it can only be prevented. It is part of the agricultural ecology. And it is dynamic, its expression can change. For example, in Serbia in 2010 it was positively associated with exposure to insecticides, with gardening, and with drinking well water and spring water [60]. In California in 2009, it was shown to be associated with consuming water from private wells in areas with documented historical pesticide use and a 70-90% increase in relative risk of Parkinson’s disease. [61]. This was not a unique observation and was in relation to agricultural crops. It is not unlikely that well water was the cause of pesticide exposure in Louisiana. In France vineyards rank among the crops that require most intense pesticide use, and regions with greater presence of vineyards are characterized with a higher Parkinson’s disease risk [62].In Israel the standardized incidence rates (SIR) for Parkinson’s disease were higher than expected in Jewish rural areas with the highest SIRs in proximity to agriculture cultivated fields and to field size [63]. In Louisiana, we have witnessed a major reduction in Parkinson’s disease incidence and distribution thanks to the introduction of glyphosate resistant crops. Now we need to firmly limit the use of 2,4-D, paraquat, and chlorpyrifos to prevent further cases.